# Co-transcriptional splicing facilitates transcription of gigantic genes

**Jaclyn M. Fingerhut** [1,2]*, **Romain Lannes** [1], **Troy W. Whitfield** [1], **Prathapan Thiru** [1], **Yukiko M. Yamashita** [1,2,3]*

1 Whitehead Institute for Biomedical Research, Cambridge, Massachusetts, United States of America, 2 Howard Hughes Medical Institute, Cambridge, Massachusetts, United States of America, 3 Department of Biology, Massachusetts Institute of Technology, Cambridge, Massachusetts, United States of America

* jaclynmf@wi.mit.edu (JMF); yukikomy@wi.mit.edu (YMY)

## Abstract

Although introns are typically tens to thousands of nucleotides, there are notable exceptions. In flies as well as humans, a small number of genes contain introns that are more than 1000 times larger than typical introns, exceeding hundreds of kilobases (kb) to megabases (Mb). It remains unknown why gigantic introns exist and how cells overcome the challenges associated with their transcription and RNA processing. The *Drosophila* Y chromosome contains some of the largest genes identified to date: multiple genes exceed 4Mb, with introns accounting for over 99% of the gene span. Here we demonstrate that co-transcriptional splicing of these gigantic Y-linked genes is important to ensure successful transcription: perturbation of splicing led to the attenuation of transcription, leading to a failure to produce mature mRNA. Cytologically, defective splicing of the Y-linked gigantic genes resulted in disorganization of transcripts within the nucleus suggestive of entanglement of transcripts, likely resulting from unspliced long RNAs. We propose that co-transcriptional splicing maintains the length of nascent transcripts of gigantic genes under a critical threshold, preventing their entanglement and ensuring proper gene expression. Our study reveals a novel biological significance of co-transcriptional splicing.

## Author summary

A small number of genes in the genome are characterized by excessively large introns, sometimes exceeding megabases. It remains poorly understood why such large introns exist or how they are processed during gene expression. By examining genes on *Drosophila* Y chromosome that contain gigantic introns, it is shown that the transcripts of these genes are co-transcriptionally spliced, and co-transcriptional splicing of these genes is critical for proper gene expression. Perturbation of splicing led to attenuation of transcription specifically in genes with gigantic introns, but not genes with average-sized introns. Whereas transcripts of genes with gigantic introns normally form separate nuclear domains, splicing perturbation led to intermingling of these domains, leading to a proposal that co-transcriptional splicing may function to keep transcript length short to avoid entanglement of transcripts, which may stall the progression of RNA polymerase II.

**Data Availability Statement:** All relevant data are within the paper and its Supporting information files. Raw sequencing reads in the form of FASTQ files, and the associated gene count matrix are available on the Gene Expression Omnibus (GEO)

under accession number GSE268126. Code for the bioinformatics analyses and intermediate files are available on GitHub at https://github.com/rLannes/Fingerhut_2024/.

**Funding:** Howard Hughes Medical Institute (YY). YY received salary from Howard Hughes Medical Institute. JF received salary from the funding to YY received from Howard Hughes Medical Institute. The funders had no role in study design, data collection and analysis, decision to publish, or preparation of the manuscript.

**Competing interests:** The authors have declared that no competing interests exist.

Taken together, these results reveal a novel function of co-transcriptional splicing that specifically impacts the expression of genes with gigantic introns.

## Introduction

Splicing, the process critical for the production of functional mRNAs through the removal of introns, is an important regulatory step during gene expression, requiring the highly coordinated actions of many splicing factors [1]. The majority of splicing occurs co-transcriptionally while the pre-mRNA is associated with the actively transcribing RNA polymerase II enzyme [2,3]. The importance of co-transcriptional splicing in proper gene expression has been extensively explored, leading to the discovery of its roles in splicing accuracy, alternative splicing, and 3' end cleavage [4–6], however other functions may yet remain to be discovered.

While introns typically range from tens of bases to several kilobases [7,8], there are notable outliers. These outliers include the introns of the human Dystrophin gene, which contains multiple introns over 100kb, making it one of the largest genes in the human genome (>2Mb gene span with only 11kb coding sequence) [9,10]. This unusual gene structure is also found in several of the *Drosophila melanogaster* Y chromosome genes (hereafter referred to as the Y-linked gigantic genes), many of which are essential for male fertility (i.e., *kl-5*, *kl-3*, *kl-2*, *ORY*, and *CCY*) (Fig 1A) [11–18].

It remains a mystery why gigantic introns exist, despite the challenges that they likely pose. Although it has been suggested that large intron size can function as a developmental timer by allowing gene expression only when cell cycle lengthens, genes in these examples are around 10–80 kb, far smaller than genes with 'gigantic' introns [19,20]. Splicing becomes more difficult as intron size increases [21–24]. Additionally, the repetitive DNAs (*e.g.*: satellite DNAs) found within the gigantic introns of the mammalian Dystrophin gene as well as the *Drosophila* Y-linked gigantic genes also likely interfere with the speed and processivity of RNA polymerases [25,26]. Moreover, the cell must invest extensive resources at every step of gene expression, including synthesizing and then degrading megabases of intronic RNAs [27,28]. Accordingly, the significance of gigantic introns remains enigmatic. Interestingly, however, between the human and mouse Dystrophin genes, intron size, but not DNA sequence, is conserved [9]. Likewise, the Y-linked gigantic genes have a relatively conserved intron-exon structure/size, yet the intronic repetitive DNAs differ even between the closely related species within the *melanogaster* subgroup (*D. melanogaster*, *D. simulans*, *D. mauritiana*, and *D. sechellia*) [29–31]. These observations raise the possibility that large intron size may have unknown functionality.

In this study, using the *Drosophila* Y-linked gigantic genes as a model to study regulation of gene expression of gigantic genes, we show that the Y-linked gigantic genes are co-transcriptionally spliced, and that this co-transcriptional splicing is critical for the progression of transcription. Whereas RNAi-mediated knockdown of splicing factors (*U2af38* and <u>Serine Arginine Protein Kinase</u> (*SRPK*)) led to global splicing defects, the Y-linked gigantic genes exhibited unique defects, where transcription attenuates toward the 3' end of the gene. Cytological examination implies that gigantic gene transcripts become entangled in the nucleus in the absence of co-transcriptional splicing, and these intertwined long transcripts may interfere with the progression of the transcription. Our study provides a novel example of the *in vivo* functionality of co-transcriptional splicing.

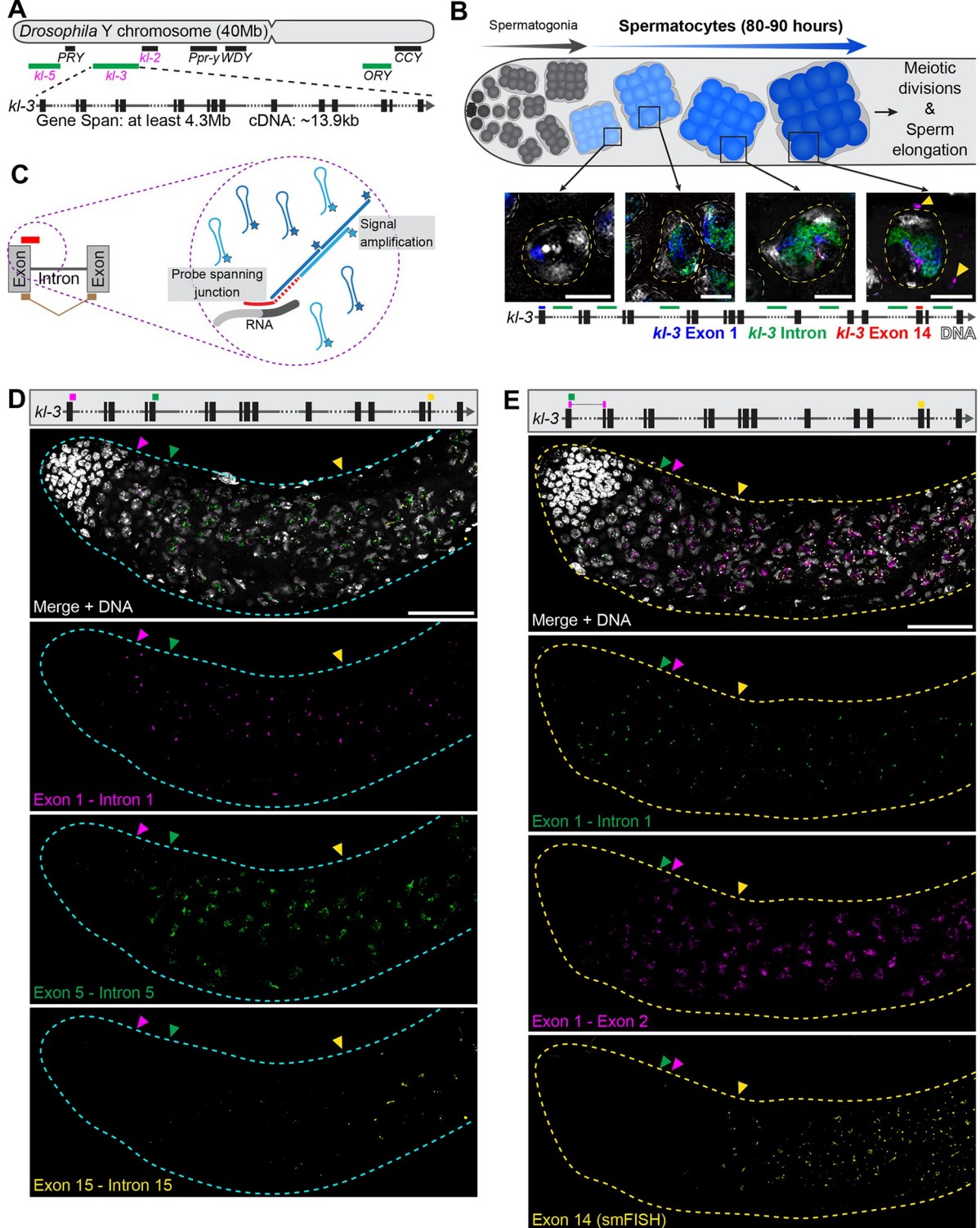

**Fig 1. The Y-linked gigantic genes are co-transcriptionally spliced.** (**A**) Diagram of the *Drosophila* Y chromosome showing locations of the Y-linked gigantic genes. Kl-granule enriched mRNAs (magenta text), Y-loop forming genes (green bars). Enlarged is a diagram of the *kl-3* gene. Exons (vertical rectangles), introns (gray line), intronic repetitive DNAs (dashed line). (**B**) Top: Diagram of *Drosophila* spermatogenesis: Early germ cells (gray) reside at one end of the testis and undergo several rounds of mitotic divisions before becoming SCs (blue). SCs develop over 80–90 h (depicted by darkening of the blue color) before initiating the meiotic divisions. Middle: RNA FISH visualizing expression of the Y-linked gigantic gene *kl-3* in single SC nuclei (yellow dashed line) at different stages of SC development. Early

exon (blue), intron (green), late exon (red), DAPI (white), nuclei of neighboring SCs (white dashed line), and cytoplasmic mRNA granules (yellow arrowheads). Bar: 10μm. Bottom: Diagram of the *kl-3* gene showing regions targeted by RNA FISH probes (colored bars). (**C**) Diagram of HCR RNA FISH. Short (~30bp) probes spanning exon–intron (red bar) or exon–exon (tan) junctions are connected to an initiator sequence (dashed line), which associates with the fluorescently tagged amplifier hairpins (blue) to initiate signal amplification. (**D**) Top: *kl-3* gene diagram showing probe target locations. Bottom: HCR RNA FISH in wildtype testes (cyan outline). Exon 1 –intron 1 (magenta), exon 5 –intron 5 (green), exon 15 –intron 15 (yellow), DNA (white). Colored arrowheads indicate earliest detection of each probe. Bar: 50μm. (**E**) Top: *kl-3* gene diagram showing probe target locations. Bottom: HCR RNA FISH in wildtype testes (yellow outline). Exon 1 – intron 1 (green), exon 1 –exon 2 (magenta), exon 14 (smFISH, yellow), DNA (white). Colored arrowheads indicate earliest detection of each probe. Bar: 50μm.

## Results

### The Y-linked gigantic genes are co-transcriptionally spliced

The *Drosophila* Y-linked gigantic genes are expressed over the course of spermatocyte (SC) development (~3.5 days [32]). In our previous study, we showed that transcription of these genes appears to progress in a 5' ➜ 3' order, taking the entirety of SC development spanning ~3.5 days: we observed that the first exon is transcribed in early SCs, followed by the transcription of intronic satellite DNA in slightly more mature SCs, then by the transcription of later exons in much later SCs (i.e. ~3 days later) (Fig 1B) [33]. This, as well as early EM studies, suggested that these gigantic genes are transcribed as singular long transcripts as opposed to trans-spliced, in which a functional mRNA is assembled from multiple RNA molecules, as has been shown for some genes with larger introns [34–37].

To gain better insights into the dynamics of Y-linked gigantic gene expression, we leveraged hybridization chain reaction (HCR) RNA FISH to allow for the visualization of various transcription products (see Methods) [38]. HCR RNA FISH allows for the detection of short target sequences (down to 20–30nt) through signal amplification, enabling the design of probes that can differentiate unspliced (nascent) vs. spliced transcripts (Fig 1C), allowing us to analyze transcription and splicing events *in vivo* and in their native context. Using probes for multiple exon–intron (nascent) junctions (exon 1 –intron 1, exon 5 –intron 5, and exon 15 –intron 15), we corroborated that transcription proceeds in 5' ➜ 3' order, as previously suggested (Fig 1D) [33]. These results, together with the fact that a large number of intronic satellite transcripts are also detected by RNA FISH (Fig 1B), are consistent with the notion that *kl-3* is likely transcribed as a single transcript. Similar results were obtained using multiple nascent transcript probes for *kl-5*, suggesting that this is a general feature of Y-linked gigantic gene transcription (S1A Fig).

Previous RNA sequencing studies as well as a few studies using *in vivo* reporter assays have suggested that the majority of genes are co-transcriptionally spliced [2,3,39–41]. Using HCR RNA FISH probes that differentiate spliced vs. unspliced transcripts (i.e. nascent exon1 -intron1 vs. spliced exon1 –exon2 RNAs), we demonstrate that *kl-3* is co-transcriptionally spliced (Fig 1E). Spliced exon 1 –exon 2 junctions were observed in early SCs, soon after the emergence of nascent exon 1 –intron 1 junctions (Fig 1E, magenta and green arrowheads). Strikingly, probes targeting exon 14 were not observed until much later in SC development, in more mature SCs (Fig 1E, yellow arrowhead). These results show that splicing of exon 1 and exon 2 occurs before exon 14 is transcribed, demonstrating the co-transcriptional splicing of *kl-3*. Likewise, the spliced product of exon 3 and exon 4 of *kl-3* was also detected before any exon 14 transcripts, further supporting that *kl-3* is co-transcriptionally spliced (S1B Fig). *kl-5* also exhibited a similar pattern: the nascent exon 1 –intron 1 junctions and spliced exon 1 –exon 2 junctions were both observed in early SCs, long before later exons (exons 16 and 17) were detected (S1C Fig). These results provide a striking example of co-transcriptional splicing, where the early introns of the

Y-linked gigantic genes are spliced days before the later exons are transcribed during cellular differentiation. HCR RNA FISH using probes against multiple spliced exon-exon junctions of *kl-3* (exon 1 –exon 2, exon 5 –exon 6, exon 11 –exon 12, exon 15 –exon 16) showed that spliced products arose in 5' ➔ 3' order over the course of SC development (S2 Fig), mirroring the pattern seen with the nascent probes (Fig 1D), and furthering the notion that splicing of the Y-linked gigantic genes occurs concordant with transcription.

## Depletion of splicing factors leads to sterility and *kl-3* splicing defects

The above results prompted us to examine whether co-transcriptional splicing plays a role in the expression of the Y-linked gigantic genes. To start to address this question, we performed RNAi mediated knockdown of key splicing factors. We selected U2af38 and SRPK, which play critical roles in splicing and are expressed in the *Drosophila* testis [42,43]. U2af38, a component of the U2AF complex, binds near the 3' splice site to facilitate spliceosome assembly [44]. SRPK phosphorylates RS domains, which are commonly found in proteins involved in splicing, to regulate their localization and/or function [45]. RNAi-mediated knockdown of these genes was performed using the *bam-GAL4* driver (*bam-gal4>UAS-RNAi*) to specifically deplete the gene products in late spermatogonia and SCs, avoiding perturbation of early germ cell development. The efficiency of knockdown was confirmed by loss of GFP signal from transgenic strains (*U2af38$^{fTRG00747.sfGFP-TVPTBF}$* and *SRPK$^{MI06550-GFSTF.1}$*) (S3A and S3B Fig) and further substantiated by RNA sequencing (see below). RNAi of *U2af38* (*bam>U2af38$^{HMS04505}$* or *bam>U2af38$^{KK108210}$*) or *SRPK* (*bam>SRPK$^{HMS04507}$* or *bam>SRPK$^{HMS04491}$*) resulted in male sterility with testes lacking mature sperm in the seminal vesicles (S4A Fig). In *SRPK* RNAi, germ cell development appeared morphologically normal at all stages (S4B Fig), but sperm were immotile and failed to exit the testis into the seminal vesicle. *U2af38* RNAi resulted in a more severe phenotype where germ cells arrested as SCs: SC development proceeds but SCs do not increase in volume to the extent seen in controls and eventually arrest prior to entry into the meiotic divisions (S4B Fig). For each gene, two independent RNAi lines resulted in the same phenotype, thus we selected one to use in all further experiments.

Knockdown of *U2af38* resulted in *kl-3* splicing defects. Using HCR RNA FISH, nascent exon 1 –intron 1 junctions and spliced exon 1 –exon 2 junctions were clearly detected in the same SC in the control (Fig 2A), consistent with *kl-3* being co-transcriptionally spliced, as shown in Figs 1 and S1. Interestingly, intron 1 –exon 2 junctions were rarely observed in control SCs (Fig 2A). This is likely because the 3' intron junction (intron–exon junction) is spliced as soon as the 3' splice site becomes available, generating the exon 1- exon2 spliced product. In contrast, in *U2af38* RNAi, the exon 1 –exon 2 spliced product was barely detectable (Fig 2A), and instead, the intron 1 –exon 2 junction was robustly detected (Fig 2A). These results indicate that splicing between exon 1 and exon 2 is defective following RNAi of *U2af38*, which led to accumulation of unspliced intron 1 –exon 2 junctions. Similar results were observed for the exon 3 –intron 3, intron 3 –exon 4, and exon 3 –exon 4 junctions of *kl-3* (Fig 2B). These results reveal the dynamics of gene expression of Y-linked gigantic genes, where the gene is transcribed in 5' ➔ 3' order and co-transcriptionally spliced as soon as the splicing acceptor site becomes available. *SRPK* RNAi also resulted in splicing defects in the Y-linked gigantic genes, which were detectable by RNA sequencing (see below for more details).

## Splicing factor depletion results in loss of mRNAs of Y-linked gigantic genes

We have previously shown that functional *kl-3*, *kl-5*, and *kl-2* mRNAs are stored in cytoplasmic ribonucleoprotein (RNP) granules, termed 'kl-granules' [46]. Kl-granules are marked by the

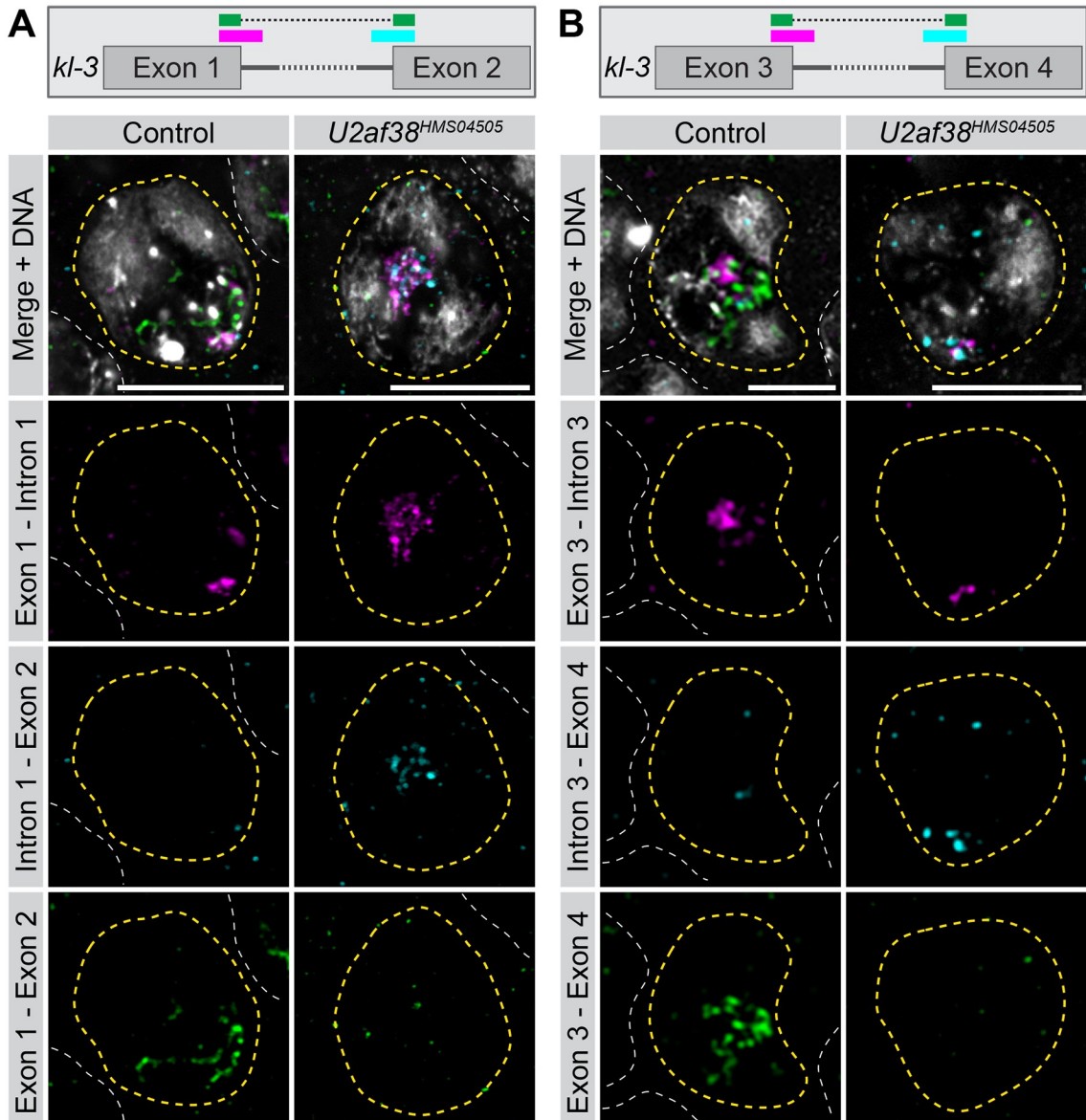

**Fig 2.** *U2af38* **RNAi results in** *kl-3* **splicing defects.** (**A** and **B**) HCR RNA FISH in control and *U2af38* RNAi SC nuclei (yellow dashed line) for the junctions shown in the *kl-3* gene diagrams. Neighboring SC nuclei (white dashed line). Bars: 10μm. (**A**) exon 1 – intron 1 (magenta), intron 1 –exon 2 (cyan), exon 1 –exon 2 (green), DNA (white). (**B**) exon 3 –intron 3 (magenta), intron 3 –exon 4 (cyan), exon 3 –exon 4 (green), DNA (white).

protein Pontin [46]. *kl-3*, *kl-5*, and *kl-2* encode axonemal dynein motor proteins and are thus essential for sperm motility [47–50]. Kl-granules containing these mRNAs assemble in SCs and are subsequently localized to the elongating sperm tail, facilitating proper incorporation of axonemal dynein proteins into the sperm flagella [46]. Kl-granule formation in SCs reflects the successful transcription and processing of *kl-3*, *kl-5*, and *kl-2* [33]. In controls, kl-granules are detected in late SCs. In marked contrast, in both *U2af38* and *SRPK* RNAi SCs, there were no kl-granules marked by Pontin protein, indicating that *kl-3*, *kl-5*, and *kl-2* might not be properly expressed/processed in these mutants (Fig 3A). Indeed, we found that component mRNAs (*kl-*

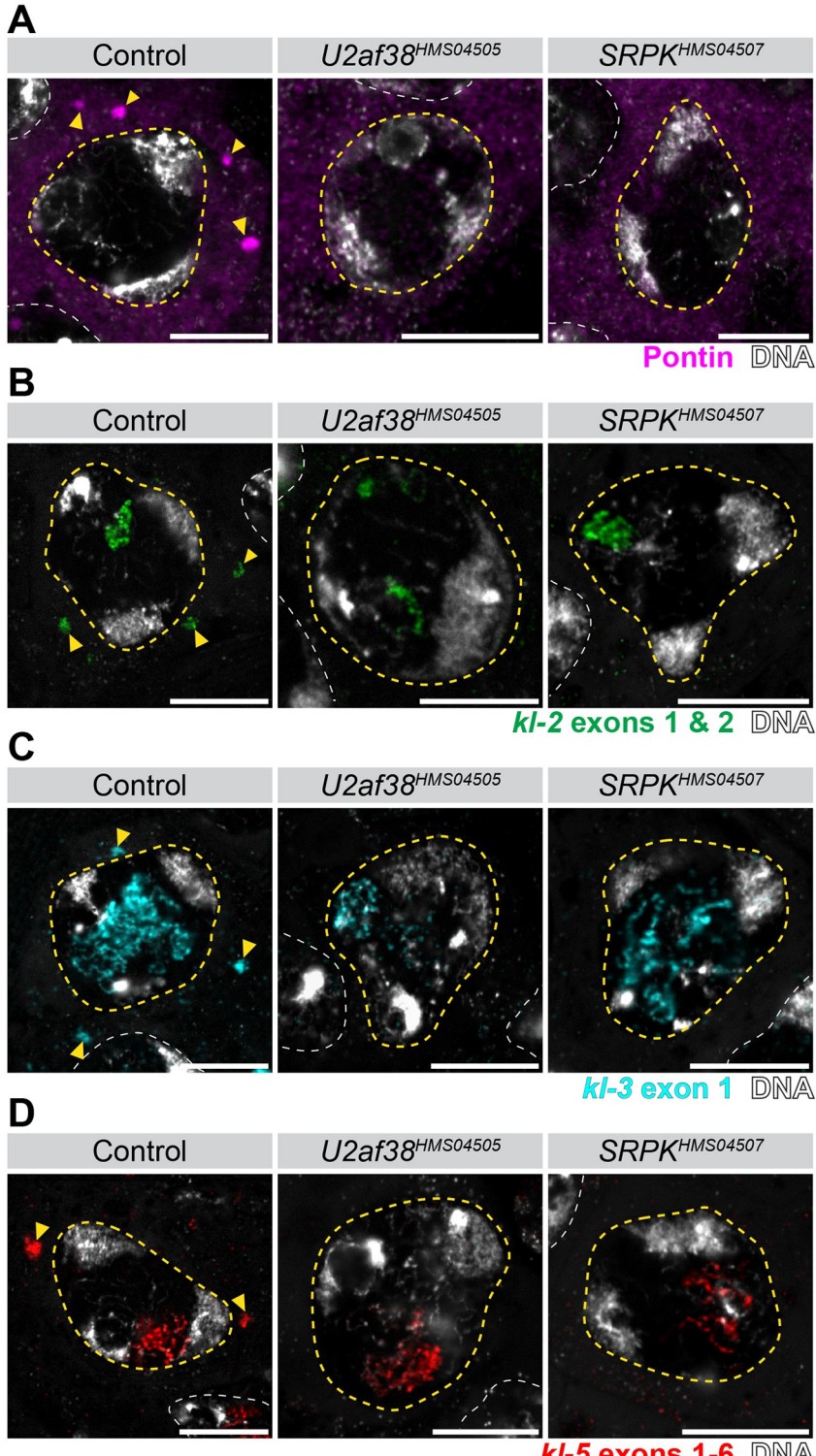

**Fig 3. kl-granules are absent in SCs upon RNAi-mediated knockdown of *U2af38* and *SRPK*. (A)** Pontin antibody staining (magenta) of single SCs (nucleus outlined in yellow dashed line, neighboring nuclei outlined with white dashed lines) in the indicated genotypes. Pontin granules (kl-granules, yellow arrowhead), DNA (white). Bars: 10μm. (**B**) *kl-2* smFISH (green) of single SCs (nucleus outlined in yellow dashed line, neighboring nuclei outlined with white dashed lines) in the indicated genotypes. *kl-2* mRNA in granules (kl-granules, yellow arrowhead), DNA (white). Bars: 10μm. (**C**) *kl-3* smFISH (cyan) of single SCs (nucleus outlined in yellow dashed line, neighboring nuclei outlined with

white dashed lines) in the indicated genotypes. *kl-3* mRNA in granules (kl-granules, yellow arrowhead), DNA (white). Bars: 10µm. (**D**) *kl-5* smFISH (red) of single SCs (nucleus outlined in yellow dashed line, neighboring nuclei outlined with white dashed lines) in the indicated genotypes. *kl-5* mRNA in granules (kl-granules, yellow arrowhead), DNA (white). Bars: 10µm. For b–d note that nuclear RNA is present in all conditions but cytoplasmic mRNAs are absent in the RNAi conditions.

*3*, *kl-5*, and *kl-2*) were not detectable in the cytoplasm of *U2af38* and *SRPK* RNAi SCs as kl-granules (Fig 3B–3D). These results suggest that mature *kl-3*, *kl-5*, and *kl-2* mRNAs are absent following RNAi-mediated knockdown of *U2af38* and *SRPK*, suggesting a failure in the expression/RNA processing of these Y-linked gigantic genes.

## Y-linked gigantic genes are downregulated upon splicing perturbation

The lack of kl-granules and component mRNAs in *U2af38* and *SRPK* RNAi SCs can be explained simply by defective splicing, which may perturb production of mRNA, mRNA export to the cytoplasm, and/or degradation of transcripts due to mis-splicing [51–53]. However, interestingly, we observed a reduction in the signal intensity of probes targeting the later exons of *kl-3* and *kl-5* within the nucleus following *U2af38* and *SRPK* RNAi (Fig 4A and 4B), suggesting that transcription itself may also be compromised. Importantly, the signal from probes targeting early exons and intronic repeats appeared unchanged in these mutants, serving as an internal control. These results suggest that the lack of kl-granules may result from a problem in transcription in addition to errors in splicing.

To gain a more quantitative assessment of the expression/splicing defects following *U2af38* and *SRPK* RNAi, we performed total RNA sequencing, comparing these RNAi conditions with controls (see Methods). We first confirmed that knockdown of *SRPK* and *U2af38* indeed resulted in a widespread disruption of splicing. Using JUM, an annotation-free method to analyze splicing errors (see Methods) [54], we found that *U2af38* and *SRPK* RNAi had strikingly different splicing patterns compared to controls, suggestive of splicing defects (S5A–S5C Fig). *U2af38* RNAi had a broader and greater number of altered splicing events than *SRPK* RNAi, consistent with the more severe cytological phenotype observed upon *U2af38* knockdown (S4B Fig). In total, 4569 genes had altered splicing events in *U2af38* RNAi and 1420 in *SRPK* RNAi ($q < 0.05$). As ~6000 genes are expressed in SCs [55], *U2af38* and *SRPK* RNAi indeed result in a large-scale disruption to the splicing in SCs. Intriguingly, we observed a strong positive correlation between splicing defects and the intron proportion (percentage of the gene span attributed to intronic sequence) in *U2af38* and *SRPK* RNAi (S5D Fig, linear model $p = 4.9 \times 10^{-04}$ for *U2af38* RNAi, $p = 5.6 \times 10^{-07}$ for *SRPK* RNAi), consistent with previous studies that indicated that large introns are more difficult to properly splice [21–24].

Differential expression analysis revealed that the Y-linked gigantic genes are specifically downregulated following *U2af38* and *SRPK* RNAi (Fig 4C and 4D). Notably, the Y-linked gigantic genes were amongst the most downregulated genes in *SRPK* RNAi and strongly downregulated in *U2af38* RNAi. When the degree of differential gene expression was analyzed with regard to intron size, genes with regular size introns (<100bp) or large (but still smaller than Y-linked gigantic genes) introns (50–100kb) did not show consistent downregulation in *U2af38* and *SRPK* RNAi, whereas the Y-linked gigantic genes exhibited clear downregulation (Fig 4E and 4F). This is not because genes with smaller introns are not aberrantly spliced: even when we selected only aberrantly spliced genes (from the analyses presented in S5 Fig), genes with intron sizes of <100bp or 50–100kb still did not show consistent downregulation (Fig 4G and 4H). Taken together, these results suggest that the loss of splicing factors has a distinct impact on the expression of the Y-linked gigantic genes. There are three autosomal genes with

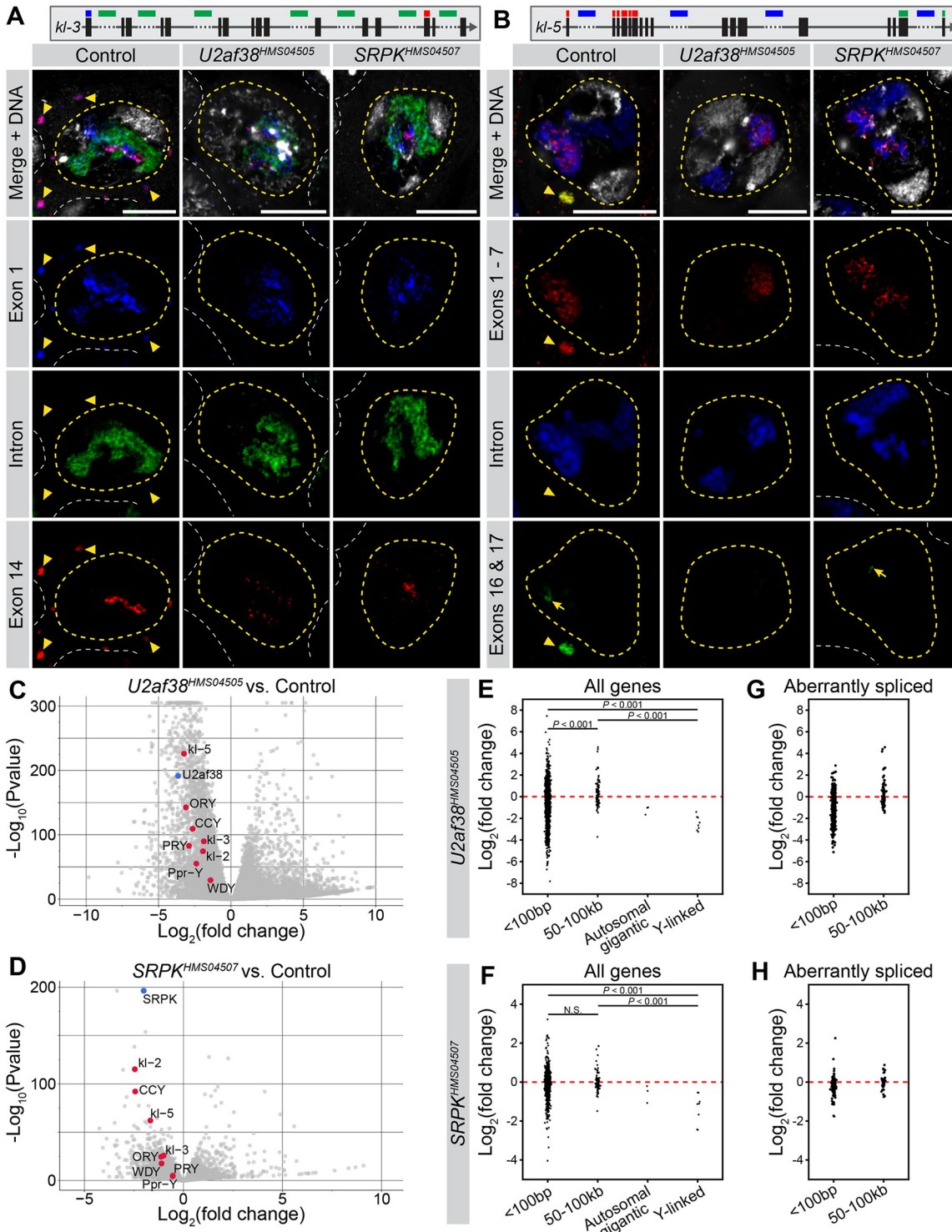

**Fig 4. Y-linked gigantic genes are downregulated upon RNAi-mediated knockdown of *U2af38* and *SRPK*. (A** and **B)** *kl-3* RNA FISH (**A**) and *kl-5* RNA FISH (**B**) in single SCs (nuclei–yellow dashed line, neighboring SC nuclei–white dashed line) in the indicated genotypes. Gene diagrams at the top indicate regions targeted by RNA FISH probes. Kl-granules (yellow arrowhead). Bars: 10μm. (**A**) *kl-3* exon 1 (smFISH, blue), intronic AATAT repeats (green), exon 14 (smFISH, red), DNA (white). (**B**) *kl-5* exons 1–7 (smFISH, red), intronic AAGAC repeats (blue), exons 16–17 (smFISH, green), DNA (white). (**C** and **D**) Volcano plots for *U2af38* RNAi (**C**) and *SRPK* RNAi (**D**) highlighting the Y-linked gigantic genes (red dots). Note that *U2af38* and *SRPK* (blue dots) are significantly downregulated, respectively. (**E–H**) Graphs showing the log₂(fold change) for all genes (**E** and **F**) or only aberrantly spliced genes (**G** and **H**) in each intron size bin in either *U2af38* (**E** and **G**) or *SRPK* (**F** and **H**) RNAi. P-values in **E** and **F** computed by Mann-Whitney U test.

gigantic introns expressed in the testis that are comparable in size to the Y-linked gigantic genes (*Pzl*, *Myo81F*, and *Mitf*: their introns also contain repetitive DNAs and there are gaps in the genome assembly) [56]. Although the results of only three genes cannot be interpreted with confidence, these genes appeared to be less impacted compared to Y-linked gigantic genes (Fig 4E–4H) (see Discussion).

## Transcription of the Y-linked gigantic genes attenuates upon splicing disruption

To gain further insight into the relationship between the splicing and transcription of the Y-linked gigantic genes, we examined individual genes' RNA sequencing reads in more detail. By plotting the read depth across the gene body for the Y-linked gigantic genes, an interesting pattern emerged. First, in control, the read depth often dropped at exons that follow gigantic (i.e. repetitive DNA-rich, gap-containing) introns (Fig 5A and 5B). Because the testis contains SCs at multiple developmental stages, this drop in sequencing depth may represent early SC populations that have not progressed to the point of transcribing later exons. Alternatively, this drop may be due, in part, to challenges in transcribing these introns, which could lead to premature dissociation of the polymerase [57]. In *U2af38* and *SRPK* RNAi, we found that the magnitude (fold change) of the read-depth drop that occurs after a gigantic intron is exacerbated compared to controls (Fig 5A and 5B, arrowheads show 'before' and 'after' the gigantic intron, fold change in *kl-3*: control = 4.3, *U2af38* RNAi = 8.2, *SRPK* RNAi = 8.3; fold change in *kl-2*: control = 1.8, *U2af38* RNAi = 8.9, *SRPK* RNAi = 2.1). Overall, the read depth across the gene span decreased clearly in *U2af38* and *SRPK* RNAi relative to the control (Fig 5C and 5D). Indeed, the RNA FISH results presented above showed a reduction in RNA FISH signal for later exons (Fig 4A and 4B). These mutant SCs have similar levels of signal from probes targeting early exons as controls (Figs 4A, 4B, 5A and 5B). Therefore, the drop in read depth in *U2af38* and *SRPK* RNAi implies attenuation of transcription along the gene length.

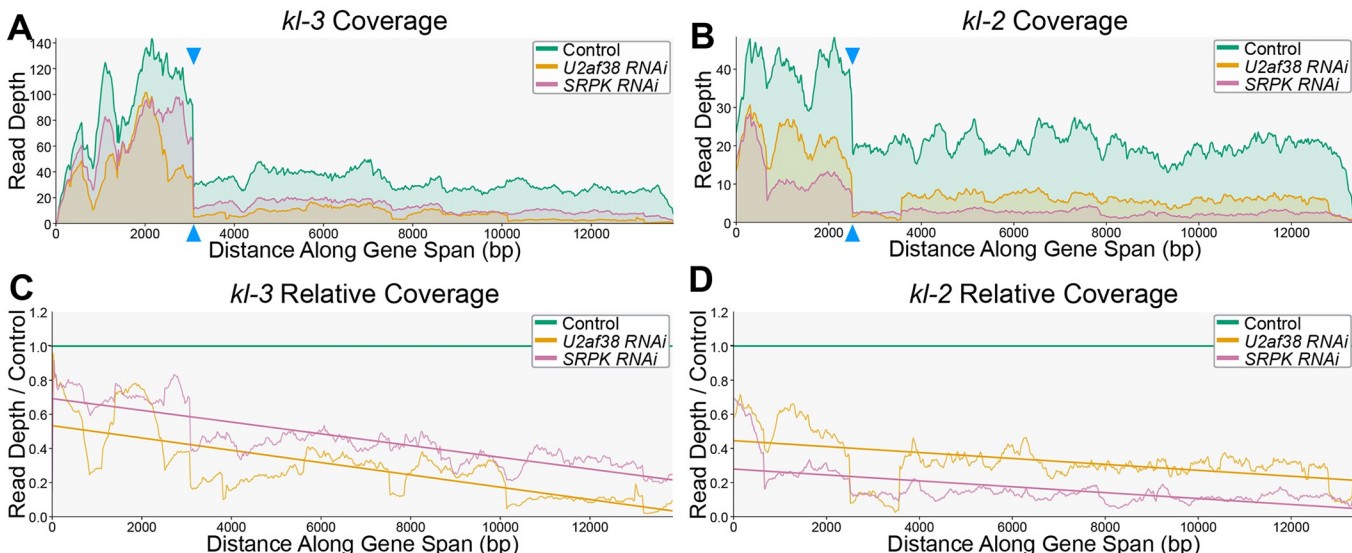

**Fig 5. Transcription of the Y-linked gigantic genes attenuates upon RNAi-mediated knockdown of *U2af38* and *SRPK*.** (**A** and **B**) Coverage plots of *kl-3* (**A**) and *kl-2* (**B**) showing the normalized read depth along the gene span (exons only) in the indicated genotypes. Blue arrowheads mark the drop in read depth that follows a gigantic intron (**C** and **D**) Plots showing the coverage along the gene span (exons only) relative to the control condition for *kl-3* (**C**) and *kl-2* (**D**) in the indicated genotypes. Linear best fit lines with negative slope indicate decreases in expression relative to the control for the RNAi conditions over the course of the gene span (5' ➔ 3').

Attenuation was observed for many of the Y-linked gigantic genes (S6 Fig). As an important comparison, transcription of genes with average size introns (max intron size 100bp or 50–100kb) did not attenuate, despite these genes being aberrantly spliced in *U2af38* RNAi and/or *SRPK* RNAi (S6 Fig). Also, attenuation was not consistently observed for autosomal genes with gigantic introns (S6 Fig, see Discussion). Taken together, these results suggest that the transcription of the Y-linked gigantic genes is particularly sensitive to splicing perturbation with splicing defects resulting in transcription attenuation.

## Transcripts of the Y-linked gigantic genes become entangled in splicing mutants

How do splicing defects lead to attenuation of transcription? Considering the gene size, it could be challenging for RNA polymerase to progress along the gene span if it is associated with a very long (megabases) unspliced pre-mRNA, and co-transcriptional splicing may be particularly critical for the Y-linked gigantic genes to minimize the length of this pre-mRNA. It is known that the transcription of the Y-linked gigantic genes *kl-3*, *kl-5* and *ORY* results in the formation of large lampbrush chromosome-like structures called 'Y-loops' (Fig 6A, control) [12]. Each gene forms a distinct Y-loop (termed Y-loops A, B, and C) that occupies a distinct, non-overlapping, domain within the SC nucleoplasm (Fig 6A, control) [12]. Interestingly, cytological examination revealed that the Y-loops intermingle with each other in *U2af38* and *SRPK* RNAi SCs (Fig 6A). Together, these results suggest that co-transcriptional splicing may aid in the efficient transcription of the Y-linked gigantic genes by preventing pre-mRNAs from becoming too long, entangling, and attenuating transcription.

## Discussion

The majority of splicing occurs co-transcriptionally [2,3]. Co-transcriptional splicing has been implicated in the regulation of RNA processing including modulating alternative splicing and 3' end cleavage [4–6]. In this study, we found that defects in the co-transcriptional splicing of the Y-linked gigantic genes resulted in a failure to complete transcription, and therefore, a lack of mature mRNA. Cytological examination implied that the long transcripts might become tangled within the nucleus, which may explain why transcription attenuates. Based on these observations, we propose that co-transcriptional splicing of the Y-linked gigantic genes prevents transcripts attached to an actively transcribing RNA polymerase II from becoming too long and therefore ensures the proper expression of these gigantic genes (Fig 6B). As an alternative possibility, splicing defects may create cryptic polyadenylation signals, leading to transcription termination. It is also possible that splicing defects may interfere with transcription elongation in general. However, considering that splicing defects do not impact transcription of genes with average-sized introns, we favor a 'long gene-specific' mechanism of transcription attenuation, such as the tangling of excessively large transcripts, although the detailed molecular explanation of transcription attenuation requires further investigations. Interestingly, a recent study showed that splicing defects lead to premature transcription termination predominantly in long genes [58], consistent with our observation in this study. Another recent study used highly expressed, long genes to examine spatial organization of transcripts, demonstrating the impact of splicing inhibition on transcription [59]. Although this study [59] suggested that their observation with long genes may be universally applicable to other genes, based on the observations described here, we postulate that long, highly expressed genes may have unique characteristics, different from short genes.

The need to keep transcripts short through co-transcriptional splicing may be of particular importance for the Y-linked gigantic genes due to the lampbrush-like nature of their

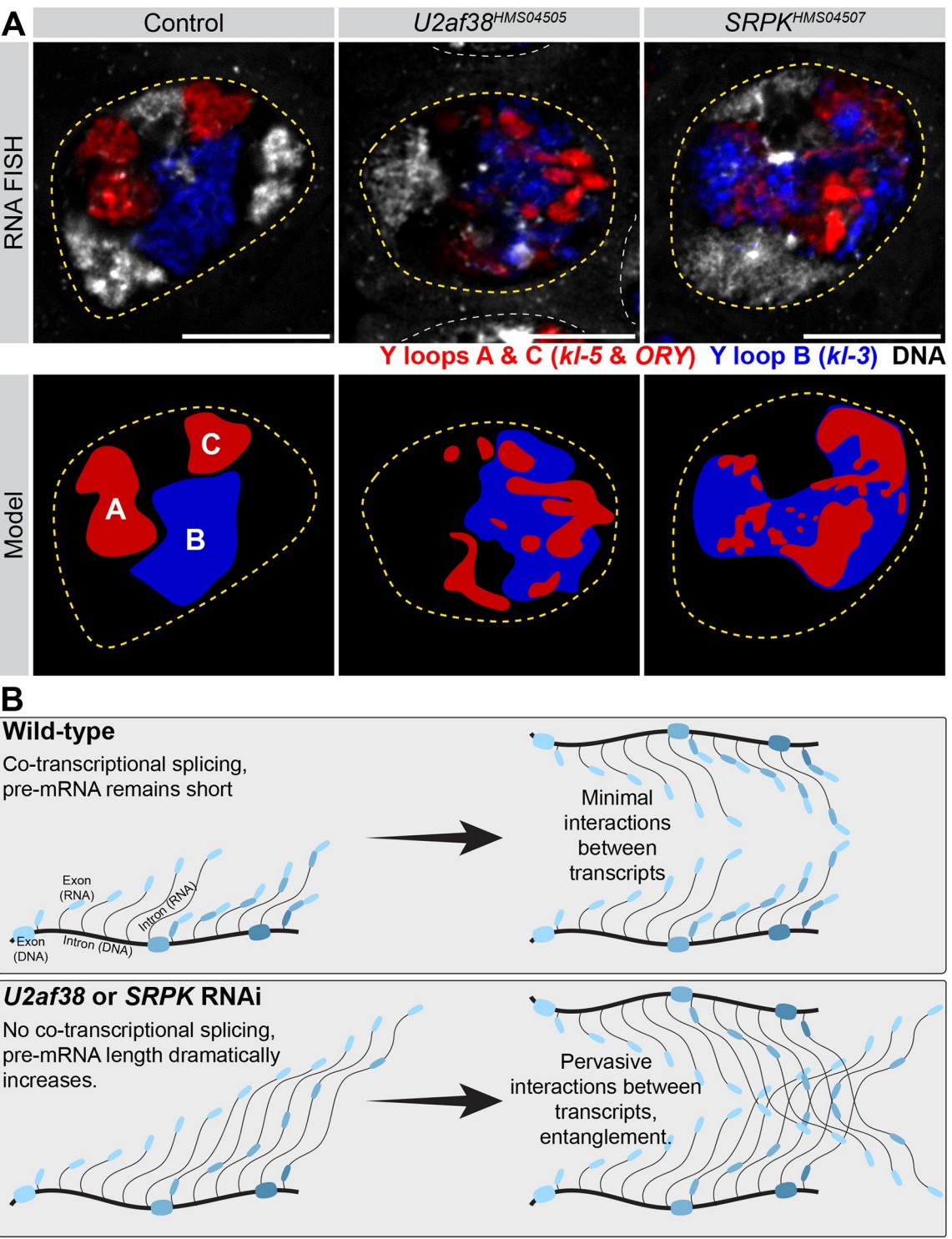

**Fig 6. Transcripts of the Y-linked gigantic genes become entangled upon RNAi-mediated knockdown of *U2af38* and *SRPK*. (A)** Top: RNA FISH for the Y-loop gene intronic transcripts in single SC nuclei (yellow dashed lines, neighboring SC nuclei, white dashed lines) in the indicated genotypes. Y-loops A (*kl-5*) and C (*ORY*) (AAGAC repeats, red), Y-loop B (*kl-3*) (AATAT repeats, blue), DNA (white). Bars: 10μm. Bottom: Models of Y-loop entanglements in each genotype matching the above RNA FISH image. (**B**) Model of how co-transcriptional splicing may function to maintain short pre-mRNAs and prevent transcript entanglement. DNA strand (thick gray line) with exons highlighted (wide colored bars), RNA transcripts (narrow gray lines) with exons highlighted (narrow colored bars). In wildtype, introns (narrow gray lines) are removed as soon as splice sites become available keeping the pre-mRNA molecule short. In splicing factor RNAi conditions, this pre-mRNA becomes increasingly long leading to entanglements.

transcription, where many RNA polymerases are simultaneously transcribing along the template DNA in a 'beads-on-string' formation [34,35,60,61]. Lampbrush chromosomes are characterized by their 'Christmas tree-like' appearance by EM, where the DNA template is bound by many active RNA polymerases and their associated transcripts of increasing size [62–64]. When many RNA polymerases are loaded on a DNA template, preventing entanglement of transcripts (either with transcripts extending from adjacent polymerases within the same gene or from neighboring lampbrush-like loci) may become more critical (Fig 6B). By contrast, conventional (non-lampbrush chromosome) genes typically have very few RNA polymerases simultaneously on the DNA template [64,65], and perhaps they are at a lower risk of transcript entanglement. This may explain why the autosomal gigantic genes *Pzl* and *Myo81F*, are not as impacted in *U2af38* and *SRPK* RNAi SCs.

The specific impact of splicing defects on the transcription of certain genes has an interesting implication. As longer genes appear to be more sensitive to splicing perturbation, this greater sensitivity could be utilized to differentially regulate gene expression. For example, the differential expression of splicing factors in different cell types may allow for distinct gene expression patterns. In this way, a certain cell type may not express subsets of genes, such as those with gigantic introns, and this regulation could influence cellular differentiation. Such regulation could add layers to gene expression programs: in addition to promoter-based and translational gene expression regulation, distinct subsets of splicing factors could influence the gene expression profile in different cell types as well. Indeed, some splicing proteins have been characterized to preferentially impact longer introns [66]. Such differential regulation could become an essential aspect of a cellular and/or developmental program, leading to the 'functionalization' of gigantic introns. Thus, whereas the function of gigantic introns remains elusive, it is tempting to speculate that gigantic introns could contribute to differential gene expression if combined with distinct expression patterns of splicing factors.

Together, the present study reveals an important role for co-transcriptional splicing in the expression of genes with gigantic introns. Future investigation is required to explore how such unique gene regulation could be utilized in broader contexts.

## Methods

### Fly husbandry

All fly stocks were raised on standard Bloomington medium at 25°C, and young flies (1- to 3-day-old adults) were used for all experiments. Flies used for wild-type experiments were the standard lab wild-type strain yw (y$^1$w$^1$). Control flies were either the parental *bam-GAL4* stock or a sibling from the same genetic cross. The following fly stocks were used: *U2af38$^{HMS04505}$* (Bloomington *Drosophila* Stock Center [BDSC]:57585), *U2af38$^{KK108210}$* Vienna *Drosophila* Resource Center [VDRC]:v110075), *SRPK$^{HMS04507}$* (BDSC:57587), *SRPK$^{HMS04491}$* (BDSC:57295), *U2af38$^{fTRG00747.sfGFP-TVPTBF}$* (VDRC:v318649), *SRPK$^{MI06550-GFSTF.1}$* (BDSC:65332), *bam-GAL4*:VP16 (BDSC:80579, gift from Dennis McKearin). We used FlyBase (release FB2024_02) to find information on gene sequences/ functions, phenotypes, and stocks [67].

### RNA fluorescent *in situ* hybridization

**Single molecule/repetitive transcript RNA FISH.** RNA FISH was performed as previously described [68]. All solutions used were RNase free. Testes from 1–3 day old flies were dissected in 1X PBS and fixed in 4% formaldehyde in 1X PBS for 30 minutes. Testes were washed briefly in 1X PBS and permeabilized in 70% ethanol overnight at 4°C. Testes were briefly rinsed with wash buffer (2X saline-sodium citrate (SSC), 10% formamide) and then

hybridized overnight at 37°C in hybridization buffer (2X SSC, 10% dextran sulfate (sigma, D8906), 1mg/mL E. coli tRNA (sigma, R8759), 2mM Vanadyl Ribonucleoside complex (NEB S142), 0.5% BSA (Ambion, AM2618), 10% formamide). Following hybridization, samples were washed three times in wash buffer for 20 minutes each at 37°C and mounted in VECTA-SHIELD with DAPI (Vector Labs). Images were acquired using a Leica Stellaris8 confocal microscope with a 63X oil immersion objective lens (NA = 1.4) and processed using Adobe Photoshop and ImageJ software.

Fluorescently labeled probes were added to the hybridization buffer to a final concentration of 50nM (for probes targeting repetitive DNA transcripts) or 100nM (for smFISH probes targeting exons). Probes against the repetitive DNA transcripts were from Integrated DNA Technologies. Probes against *kl-3*, *kl-5*, and *kl-2* exons were designed using the Stellaris RNA FISH Probe Designer (Biosearch Technologies, Inc.) available online at www.biosearchtech.com/stellarisdesigner. Each set of custom Stellaris RNA FISH probes was labeled with Quasar 670, Quasar 570 or Fluorescein-C3. Probe information can be found in (S1 Table).

**HCR RNA FISH.** All solutions used were RNase free. Testes from 1–3 day old flies were dissected in 1X PBS and fixed in 4% formaldehyde in 1X PBS for 20 minutes and then washed twice, 5 minutes per wash, in 1X PBS 0.1% Tween-20. Permeabilization was achieved by washing for 2 hours in 1X PBS 0.1% Triton X-100. The testes were then washed twice, 5 minutes per wash, in 5X SSC 0.1% Tween-20 and blocked with 100ug/mL of salmon sperm DNA for 30 minutes at 37C in Hybridization Buffer (Molecular Instruments). Probes were added to Hybridization Buffer to a final concentration of 10nM. If combining with smFISH probes, smFISH probes were also added to Hybridization Buffer to their standard concentration (see above). Hybridized overnight at 37C shaking. Samples were washed four times, 15 minutes per wash, at 37C with Probe Wash Buffer (Molecular Instruments). During the second wash, hairpin solutions for desired amplifiers (Molecular Instruments) were created by individually heating each hairpin at 95C for 90 seconds in a PCR machine and allowing the samples to slowly cool to room temperature. Amplifiers were added to Amplification Buffer (Molecular Instruments) to a final concentration of 60nM. Samples were washed twice, 5 minutes per wash, with 5X SSC 0.1% Tween-20, and then transferred to Amplification Buffer + amplifiers and incubated overnight at room temperature nutating. Testes were then washed twice, 30 minutes per wash, with 5X SSC 0.1% Tween-20 and mounted in VECTASHIELD with DAPI (Vector Labs). Images were acquired using a Leica Stellaris8 confocal microscope with a 63X oil immersion objective lens (NA = 1.4) and processed using Adobe Photoshop and ImageJ software.

HCR probes designed to span exon-intron or exon-exon junctions were the "V2" HCR probe design [38]. Approximately 15bp on either side of the junction was selected for 30bp of target sequence. Probe information can be found in (S1 Table).

## Immunofluorescence staining

Testes were dissected in 1X PBS, transferred to 4% formaldehyde in 1X PBS, and fixed for 30 minutes. Testes were then washed in 1X PBST (PBS containing 0.1% Triton X-100) for at least 60 minutes followed by incubation with primary antibodies diluted in 1X PBST with 3% BSA at 4°C overnight. Samples were washed for at least 1 hour in 1X PBST, incubated with secondary antibody in 1X PBST with 3% BSA at 4°C overnight, washed as above, and mounted in VECTASHIELD with DAPI (Vector Labs). Images were acquired using Leica Stellaris8 confocal microscope with a 63X oil immersion objective lens (NA = 1.4) and processed using Adobe Photoshop and ImageJ software.

The primary antibody used was anti-Pont (1:200; guinea pig) [46]. Alexa Fluor–conjugated secondary antibodies (Life Technologies) were used at a dilution of 1:200.

## RNA isolation and sequencing

Total RNA was purified from adult testes (100 pairs/sample) by TRIzol (Invitrogen) extraction according to the manufacturer's instructions. Libraries were prepared for RNA sequencing using the KAPA Biosystems RNA HyperPrep Kit with RiboErase according to manufacturer's directions with some modifications. Briefly, 500 ng of total RNA was ribo-depleted by hybridization of complementary DNA oligonucleotides. The set of complementary oligonucleotides was a custom panel designed for *Drosophila*. This was followed by treatment with RNase H and DNase to remove rRNA duplexed to DNA and original DNA oligonucleotides. The enriched fraction was then fragmented with heat and magnesium, and first-strand cDNA was generated using random primers. Strand specificity was achieved during second-strand cDNA synthesis by replacing dTTP with dUTP, which quenches the second strand during amplification, and the cDNA is then A-Tailed. The final double strand cDNA was then ligated with indexed adapters. Finally, the library was amplified using a DNA Polymerase that cannot incorporate past dUTPs, effectively quenching the second strand during PCR. Libraries were enriched for fragments between 500–1000bp with two additional cycles of PCR followed by a size selection using a 1.5% gel on a Pippin Prep (Sage Science) electrophoresis instrument. Final libraries were quantified by qPCR and Fragment Analyzer. Samples were sequenced on a NOVASEQSP, producing 250 × 250bp paired-end reads.

## Bioinformatic analyses

**Differential gene expression.**   Paired-end reads (250 x 250 bp) were mapped to the *D. melanogaster* Release 6 (dm6) reference genome using the STAR aligner (v. 2.7.1a) [69]. Counts for fly protein coding genes and lncRNAs (FlyBase Dmel Release 6.37 annotations) were tabulated using featureCounts with the appropriate strand-specific setting [70]. Differential expression of mRNAs was assessed for pairwise contrasts between conditions using estimated fold-changes and the Wald statistic in DESeq2 (v. 1.36.0) [71].

**Analysis of aberrant splicing events.**   Aberrant splicing events between pairs of conditions (control vs RNAi) were detected using the Junction Usage Model (JUM) [54]. For application of JUM (v. 2.0.2), the paired-end reads were re-mapped to the *D. melanogaster* Release 6 (dm6) reference genome using the STAR aligner (v. 2.7.1a) in its two-pass mode, as suggested by the JUM developers (https://github.com/qqwang-berkeley/JUM). For each genotype, we defined a gene as aberrantly spliced if the q-value provided by JUM for the gene was below 0.05. Further analyses only looked at protein coding genes.

**Intron definitions.**   To investigate the impact of intron content for each gene on its expression we defined two values for each gene: the intron proportion and the largest intron size. The intron proportion was calculated for each gene, using the gene's largest isoform, as 1 minus the ratio of the sum of the sizes of all its exons over the gene's length. The size of the longest intron for a gene was defined as the longest intron present in any of the gene's isoforms. We calculated the distribution of the longest intron within the longest transcript for each protein-coding gene. Using this distribution, we selected a threshold (100 bp) to include the mode and genes with introns < 100bp were considered to represent "average" genes. The 50–100kb bin was selected to represent genes that are large but not yet so large to be outliers (like the Y-linked gigantic genes or the autosomal gigantic genes) in the distribution of the largest intron per transcript.

**Gene coverage.** To investigate gene coverage changes along the gene's length, we first split the BAM alignment files by the strandness of the reads using Samtools (1.11 (using htslib 1.11)). For each selected gene in each genotype, we used pysam (0.21.0) to compute the read coverage in a strand specific manner along the exons. We normalized the coverage using the total number of aligned reads multiplied by a factor of 10 million. To examine the coverage differences between mutant genotypes and the control, we calculated the ratio in the gene coverage between each mutant genotype and the control. Regression lines were fitted using the polyfit function (degree 1) from the numpy package (1.24.3). The genes from the 100bp and 50–100kb bins shown in S6 Fig were selected from the aberrantly spliced genes in each bin and further filtered by expression in the control (each selected gene was strongly expressed in the control), differential expression (downregulation with significant adjusted p-value and fold change), simplicity of the locus (few known isoforms, genome annotation matches the reads, no close neighboring genes producing downstream of gene transcripts), and linear correlation coefficient. All figures were generated using python (3.8.10), matplotlib (3.7.1) and pandas (2.0.0).

Code for the bioinformatics analyses and intermediate files are available on GitHub at https://github.com/rLannes/Fingerhut_2024/.

## Supporting information

**S1 Fig. *kl-3* and *kl-5* transcription and co-transcriptional splicing.** (**A**) Top: *kl-5* gene diagram showing probe target locations. Bottom: HCR RNA FISH in wildtype SCs of increasing maturity [single SC nuclei (yellow dashed line), neighboring SC nuclei (white dashed line). Exon 1 –intron 1 (yellow), exon 12 –intron 12 (magenta), exons 16 & 17 (smFISH, cyan), DNA (white). Bars: 10μm. (**B**) Top: *kl-3* gene diagram showing probe target locations. Bottom: HCR RNA FISH in wildtype testes (yellow dashed outline). Exon 3 –exon 4 (blue), exon 14 (smFISH, red), DNA (white). Colored arrowheads indicate earliest detection of each probe. Bar: 50μm. (**c**) Top: *kl-5* gene diagram showing probe target locations. Bottom: HCR RNA FISH in wildtype testes (yellow dashed outline). Exon 1 –intron 1 (red), exon 1 –exon 2 (blue), exons 16 & 17 (smFISH, yellow), DNA (white). Colored arrowheads indicate earliest detection of each probe. Bar: 50μm.
(TIF)

**S2 Fig. *kl-3* is spliced concordant with transcription.** (**A**) Top: *kl-3* gene diagram showing probe target locations. Bottom: HCR RNA FISH in wildtype testes (yellow dashed outline). Exon 1 –exon 2 (blue), exon 5 –exon 6 (green), exon 11 –exon 12 (magenta), exon 15 –exon 16 (yellow), DNA (white). Colored arrowheads indicate earliest detection of each probe. Bar: 50μm.
(TIF)

**S3 Fig. RNAi efficiency of *U2af38* and *SRPK*.** (**A**) U2af38-GFP expression in the apical tip of the testis (cyan dashed line) in the indicated genotypes. *bam* driven RNAi expression starts part way through spermatogonial differentiation (yellow dashed line), leaving the germline stem cells/early spermatogonia unaffected. Bars: 50μm. (**B**) SRPK-GFP expression in the apical tip of the testis (cyan dashed line) in the indicated genotypes. *bam* driven RNAi expression starts part way through spermatogonial differentiation (yellow dashed line), leaving the germline stem cells/early spermatogonia unaffected. Bars: 50μm.
(TIF)

**S4 Fig. RNAi of *U2af38* and *SRPK* results in male sterility.** (**A**) Seminal vesicles in the indicated genotypes. DNA (white). Bars: 50μm. Yellow arrows indicate round epithelial cells while

yellow arrowheads indicate needle-shaped sperm nuclei. (**B**) Whole testes (yellow dashed lines) in the indicated genotypes. DNA (white), maturing sperm nuclei (yellow arrows). Diagram (bottom left) illustrates proper germ cell development. Spermiogenesis is absent in *U2af38* RNAi testes while *SRPK* RNAi testes appear phenotypically normal.
(TIF)

**S5 Fig. RNAi-mediated knockdown of *U2af38* and *SRPK* results in broad splicing defects.** (**A** and **B**) Graphical representation of the number (**A**) and proportion (**B**) of different types of aberrant splicing events in *U2af38* and *SRPK* RNAi conditions that were detected by JUM (q < 0.05). (**C**) Total number of genes aberrantly spliced in *U2af38* and *SRPK* RNAi. (**D**) Graph showing the proportion of genes in each intron proportion bin (percent of the gene span that is intronic) that are aberrantly spliced in *U2af38* and *SRPK* RNAi.
(TIF)

**S6 Fig. Effect of RNAi-mediated knockdown of *U2af38* and *SRPK* on transcription based on intron size.** (**a**–**d**) Plots showing the coverage along the gene span (exons only, normalized) relative to the control condition for selected aberrantly spliced genes in either *U2af38* (left column) or *SRPK* (right column) RNAi. Linear best fit lines estimate changes in expression over the gene span relative to the control condition. (**a**) max intron size less than 100bp. (**b**) max intron size between 50–100kb. (**c**) Autosomal genes with gigantic introns. (**d**) The Y-linked gigantic genes. *PRY* and *WDY* omitted due to low overall expression and/or errors in the genome annotation (see Methods).
(TIF)

**S1 Table. RNA FISH probes.**
(DOCX)

**S1 Data. S5 Fig and Fig 4 numerical data.**
(XLSX)

## Acknowledgments

We thank the Bloomington Stock Center and the Vienna *Drosophila* Resource Center for reagents, the members of the Yamashita lab for discussions and comments on the manuscript and Drs. Phil Sharp, Chris Burge, Don Rio, Qingqing Wang, Stephen Smale, Douglas Black and Mariana Wolfner for helpful suggestions. We thank Drs. Ruth Lehmann, Sherilyn Grill, and Julia Wucherpfennig for discussion, sharing of reagents and assistance with HCR RNA FISH. We thank the Genome Technology Core at the Whitehead Institute for their consultation and aid in designing and performing RNA sequencing experiments.

## Author Contributions

**Conceptualization:** Jaclyn M. Fingerhut, Yukiko M. Yamashita.

**Formal analysis:** Jaclyn M. Fingerhut, Romain Lannes, Troy W. Whitfield, Prathapan Thiru.

**Funding acquisition:** Yukiko M. Yamashita.

**Investigation:** Jaclyn M. Fingerhut, Yukiko M. Yamashita.

**Supervision:** Yukiko M. Yamashita.

**Validation:** Jaclyn M. Fingerhut.

**Visualization:** Jaclyn M. Fingerhut.

**Writing – original draft:** Jaclyn M. Fingerhut, Yukiko M. Yamashita.

**Writing – review & editing:** Jaclyn M. Fingerhut, Romain Lannes, Troy W. Whitfield, Yukiko M. Yamashita.

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
