## [Decision Letter · Decision Letter 0]

20 May 2024

Dear Yukiko,

Thank you very much for submitting your Research Article entitled 'Co-transcriptional splicing facilitates transcription of gigantic genes' to PLOS Genetics.

The manuscript was fully evaluated at the editorial level and by independent peer reviewers. The reviewers appreciated the attention to an important topic but identified some concerns that we ask you address in a revised manuscript.

We therefore ask you to modify the manuscript according to the review recommendations. Your revisions should address the specific points made by each reviewer. Specifically, Review #1 strongly suggests a western blot to test U2af38 and SRPK downregulation on Pol II CTD phosphorylation. 

Yours sincerely,

Giovanni Bosco, Ph.D.

Section Editor

PLOS Genetics

Fengwei Yu

Section Editor

PLOS Genetics

Reviewer's Responses to Questions

**Comments to the Authors:**

Reviewer #1: Fingerhut and colleagues report that when splicing is inhibited, very long genes fail to be completely transcribed. Authors analyzed the Drosophila Y-linked gigantic genes that are expressed during spermatocyte development. Using RNA FISH to image spliced and unspliced transcripts, authors show that exons 1 and 2 are ligated before exon 14 is transcribed, which occurs days later. This observation is consistent with the prevailing model that splicing occurs co-transcriptionally. Next, authors used RNAi to downregulate critical splicing factors (U2af38 and SRPK) and showed that both are required for proper sperm formation. Moreover, splicing was inhibited, as expected. Most interestingly, transcripts ceased to be detected beyond the very long intronic regions, suggesting an impairment of transcription elongation. However, authors do not elucidate the mechanism underlying this phenomenon. Instead, they propose a somewhat nebulous model suggesting that transcription is hindered by the presence of tangled, unspliced long pre-mRNAs.

Notably, it has been previously shown that PlaB, a small molecule splicing inhibitor, induces premature transcription termination, predominantly in long genes (doi: 10.1016/j.molcel.2021.02.034; see Fig. 4). Treatment with PlaB was also shown to cause massive abortion of transcription of long genes by FISH (doi:10.1038/s41556-022-00847-6; see Fig. 7). Another study showed that splicing inhibition decreased phosphorylation level of Ser2 in Pol II CTD (https://doi.org/10.1093/nar/gkv740).

It is imperative for authors to contextualize their findings within the framework of these prior studies. I strongly advocate conducting a Western blot analysis to investigate the impact of U2af38 and SRPK downregulation on Pol II CTD phosphorylation. Should this analysis yield affirmative results, it suggests a plausible mechanism for the observed transcriptional attenuation: splicing-dependent phosphorylation of Pol II CTD.

Reviewer #2: This article describes a novel function for co-transcriptional splicing: keeping the pre-mRNA transcript length of gigantic genes short to ensure proper expression. The Drosophila Y chromosome provides an excellent model for studying intron gigantism – the Y chromosome encodes only 20 genes, 6 of which are giant genes with large satellite DNA rich-introns. These Y-linked gigantic genes are only expressed during spermatogenesis, take days(!) to transcribe, and act as fertility factors. The authors’ use of hybridization chain reaction (HCR) RNA FISH to detect short target sequences in spermatocytes and visualize distinct transcription products (i.e. exon-intron products for nascent mRNA vs exon-exon products for spliced transcripts) is ingenious. With this technique, the authors confirm their previous work that Y-linked gigantic genes are transcribed as a single transcript and that transcription proceeds in the 5’-3’ order. Additionally, the authors demonstrate that these genes are co-transcriptionally spliced, and that RNAi-mediated depletion of critical splicing factors U2af38 and SRPK results in downregulation of Y-linked gigantic genes and sterility. The authors propose that splicing defects may lead to attenuation of transcription by preventing mRNAs from becoming too long and getting entangled. While the question remains whether giant introns serve any functional purpose, this work hints at the possibility that giant introns may facilitate differential gene expression when combined with specific expression of splicing factors.

This manuscript is well-written, easy to read, and accompanied by beautiful images. I have only minor comments:

1. Line 105: In the text the authors mention that “kl-3 is co-transcriptionally spliced (Fig. 1E). Spliced exon 1 – exon 2 junctions were observed in early SCs alongside nascent exon 1 – intron 1 junctions (Fig. 1E, magenta and green arrowheads).” However, the data in Figure 1E shows green probe (exon 1 – intron 1) before the appearance of magenta (exon 1 – exon 2). Are both exon 1 – intron 1 junctions and spliced exon 1 – exon 2 junctions always detected in the same cells? If not, the authors should place the magenta arrow immediately to the right of the green arrow to reflect the earliest detection of each probe more accurately. This would still be consistent with their conclusion that Y-linked gigantic genes are co-transcriptionally spliced.

2. Line 546, Figure S1B: “exon3 – intron4” should be “exon 3 – exon4” to accurately reflect the probe target locations shown in the kl-3 gene diagram.

3. Figure S1B and C: It is very difficult to see the dark blue FISH probes in printed versions of the manuscript (it is only slightly clearer on a computer) – the authors should consider using another color (cyan or white?) to make visualizing the data easier.

4. Figure 4: the letter B is missing from the panel.

5. Line 196: The authors mentioned 1429 altered splicing events in SRPK RNAi – only 1420 are denoted in Figure S5C.

6. Could inhibition of splicing lead to the appearance of cryptic polyadenylation signals in unspliced giant introns and result in premature transcription termination? If so, could this partially explain the drop in read depth that follows a giant intron (but not a smaller intron which may not contain cryptic polyadenylation signals)? It would be helpful if the authors clarify this point (note: I am not requesting an experiment as it would be beyond the scope of this work; a simple discussion point would suffice here).

**Have all data underlying the figures and results presented in the manuscript been provided?**

Reviewer #1: Yes

Reviewer #2: Yes

PLOS authors have the option to publish the peer review history of their article (what does this mean?). If published, this will include your full peer review and any attached files.

Reviewer #1: No

Reviewer #2: No

---

## [Editor Report · Decision Letter 1]

31 May 2024

Dear Yukiko,

Thank you for your thorough response to the reviewers' comments. We are pleased to inform you that your manuscript entitled "Co-transcriptional splicing facilitates transcription of gigantic genes" has been editorially accepted for publication in PLOS Genetics. Congratulations!

Yours sincerely,

Giovanni Bosco, Ph.D.

Section Editor

PLOS Genetics

Fengwei Yu

Section Editor

PLOS Genetics

Comments from the reviewers (if applicable):

**Data Deposition**

http://datadryad.org/submit?journalID=pgenetics&manu=PGENETICS-D-24-00351R1

**Press Queries**

---

## [Editor Report · Acceptance letter]

6 Jun 2024

PGENETICS-D-24-00351R1 

Co-transcriptional splicing facilitates transcription of gigantic genes 

Dear Dr Yamashita, 

We are pleased to inform you that your manuscript entitled "Co-transcriptional splicing facilitates transcription of gigantic genes" has been formally accepted for publication in PLOS Genetics! Your manuscript is now with our production department and you will be notified of the publication date in due course.

With kind regards,

Lilla Horvath

PLOS Genetics

On behalf of:
